# Intranasal Immunization of Pneumococcal *pep27* Mutant Attenuates Allergic and Inflammatory Diseases by Upregulating Skin and Mucosal Tregs

**DOI:** 10.3390/vaccines12070737

**Published:** 2024-07-03

**Authors:** Hamid Iqbal, Dong-Kwon Rhee

**Affiliations:** 1Department of Pharmacy, CECOS University, Peshawar 25000, Pakistan; hamid.bukhari11@gmail.com; 2School of Pharmacy, Sungkyunkwan University, Suwon 16419, Republic of Korea

**Keywords:** Treg cells, nasal vaccine, mucosal tolerance, allergy, inflammatory diseases

## Abstract

Conventional immunization methods such as intramuscular injections lack effective mucosal protection against pathogens that enter through the mucosal surfaces. Moreover, conventional therapy often leads to adverse events and compromised immunity, followed by complicated outcomes, leading to the need to switch to other options. Thus, a need to develop safe and effective treatment with long-term beneficial outcomes to reduce the risk of relapse is mandatory. Mucosal vaccines administered across mucosal surfaces, such as the respiratory or intestinal mucosa, to prompt robust localized and systemic immune responses to prevent the public from acquiring pathogenic diseases. Mucosal immunity contains a unique immune cell milieu that selectively identify pathogens and limits the transmission and progression of mucosal diseases, such as allergic dermatitis and inflammatory bowel disease (IBD). It also offers protection from localized infection at the site of entry, enables the clearance of pathogens on mucosal surfaces, and leads to the induction of long-term immunity with the ability to shape regulatory responses. Regulatory T (Treg) cells have been a promising strategy to suppress mucosal diseases. To find advances in mucosal treatment, we investigated the therapeutic effects of intranasal *pep27* mutant immunization. Nasal immunization protects mucosal surfaces, but nasal antigen presentation appears to entail the need for an adjuvant to stimulate immunogenicity. Here, a novel method is developed to induce Tregs via intranasal immunization without an adjuvant to potentially overcome allergic diseases and gut and lung inflammation using lung–gut axis communication in animal models. The implementation of the *pep27* mutant for these therapies should be preceded by studies on Treg resilience through clinical translational studies on dietary changes.

## 1. Introduction

Mucosal vaccines administer elicit mucosal immunity at the site of administration and in certain other mucosal compartments depending on the routes of administration. Mucosal vaccines can target specific mucosal surfaces, such as respiratory, genital, or intestinal mucosa. Both nasal and sublingual vaccines can elicit mucosal immunity in the upper and lower respiratory tract, stomach, and small intestine. Oral, rectal, and vaginal vaccines induce mucosal immunity in the GI tract, colon, and rectum [1,2,3,4]. Furthermore, mucosal immunization vaccines offer several other advantages over traditional systemic vaccination by inducing higher levels of antibodies and protecting mucosal surfaces from pathogen infection [3,4]. With the advent of COVID-19, it has become even more important to block pathogens at the nasal mucosa entrance. Therefore, to prevent the colonization of respiratory pathogens, the need for nasal vaccines is a prerequisite to overcoming conventional injectable vaccines [5]. Nasal immunization is considered to be the most effective method of inducing mucosal immunity in the nasopharynx, lungs, and vagina [6], but whether or not it can stimulate skin and intestinal mucosal immunity remains unknown.

Most antigens are not sufficient to induce mucosal immunity and require immune-boosting adjuvants such as cholera toxin and heat-labile enterotoxin. However, a clinical study found that an inactivated nasal influenza vaccine containing adjuvants caused facial paralysis (Bell’s palsy) in some people [7]. Moreover, mucosal adjuvants can impair the olfactory system of mice [8]. Therefore, nasal immunization is preferred to be administered without adjuvants to avoid pathological conditions.

Current treatments for allergic dermatitis, rhinitis, asthma, and inflammatory bowel disease (IBD) do not offer a cure, but provide temporary relief; β-2 agonists and inhaled corticosteroids can be used for mild asthma symptoms, while antibodies to type 2-dependent cytokines (IL-4, IL-5, and IL-13) can be used for severe allergies [9]. However, these treatments often lead to drug resistance or side effects, resulting in a switch to other therapies [10,11]. Therefore, safe and highly effective treatments need to be developed.

Regulatory T (Treg) cells have been known to suppress inflammatory responses [12,13]. Treg cells can be utilized to maintain immune homeostasis by relieving excessive inflammation or preventing autoimmunity after a pathogenic event. In murine models, Tregs can regulate both low- and high-level inflammation caused by type 2-hypocytokine and type 2-hypercytokine secretion, respectively, and in human cells, they can alleviate allergic airway inflammation [9]. Non-toxic Treg cells on the oral and nasal mucosal surfaces are induced similarly to cells in the gut. Tolerance of intestinal Treg is induced only when low amounts of antigen are delivered, and at high antigen doses, anergy is induced [14]. Respiratory tolerance has mechanisms similar to intestinal oral tolerance mechanisms [15], but it is not known whether or not nasal and mucosal tolerance can be regulated within the nasal cavity.

## 2. Pneumococcal Pep27 Induction during Invasion and Lack of Sepsis Induction by *pep27* Mutant

*Streptococcus pneumoniae* (pneumococcus) is carried asymptomatically in the nasopharynx of healthy individuals, and this serves as a major reservoir for pneumococcal infections [16]. Pneumococcus causes various potentially life-threatening infections such as pneumonia, bacteremia (sepsis), and meningitis [17]. A prerequisite for invasive pneumonia is that pneumococci must colonize the nasopharynx before they can progress to invasive pneumonia and disseminate to the lung, bloodstream, and central nervous system [18]. In pneumococci, bacterial lysis releases cell wall components and pneumolysin toxin, and subsequently triggers pro-inflammatory responses. Moreover, mutations in the major autolysin (LytA) reduce pneumococcal virulence [19].

Pep27 is an effector molecule of the *vncRS* operon that mediates vancomycin resistance and autolysis [20,21]. However, via microarray analysis, we discovered that a number of pneumococcal genes were induced upon invasion of the human lung cell line A549. We confirmed that these target genes were indeed induced upon invasion of A549 cells via real-time PCR, which demonstrated that not only *pep27*, but also *vncR* and *vncS* were activated via pneumococcal infection. Furthermore, when we constructed 15 mutants of the genes induced during A549 invasion and tested them for attenuation of cytotoxicity in vitro after infection with A549 cells, we confirmed reduced toxicity in mice via intranasal infection (pneumonia model) or intraperitoneal injection (sepsis model). Of those genes, the *pep27* gene of the *vncRS* operon was more prominently induced than normal controls (Table 1). However, the most significantly induced gene was always *pep27*, and the *pep27* mutant was found to have the least toxicity and vaccine efficacy. The lysis-resistant *pep27* mutant (Δpep27) gives rise to reduced cytotoxicity to host cells, resulting in decreased inflammation and death [22,23]. The Δpep27 mutant could be a potentially useful mutation for use in an inexpensive live-attenuated vaccine that could be used to elicit mucosal immunity and attenuate virulence. Based on these features, we immunized the mice three times, once/week, and measured the relevant parameters for experimental purposes, as shown in Figure 1. Thus, Δ*pep27* makes the pneumococci incapable of infiltrating into the lungs, blood, and brain [18], resulting in a virtually non-cytotoxic and highly safe agent that does not cause death after injection into the brains of immunocompromised mice [24]. Furthermore, intranasal immunization with Δ*pep27*, without any adjuvant, demonstrated long-term protective efficacy [18].

Intranasal immunization with attenuated erythromycin-resistant Δpep27 and inactivated markerless Δ*pep27* could protect a host from lethal pneumococcal challenge serotype independently; it also lowers bacterial colonization in the nasopharynx [18,25], suggesting that Δpep27 may be able to provide mucosal immunity against pneumococcal diseases and could represent an efficient mucosal vaccine. Additionally, Δpep27 immunization protected against heterologous strains in bronchoalveolar lavage fluid at the nasopharynx, suggesting that Δpep27 immunization provides a wide range of cross-protection, demonstrating long-lasting immunity [27]. Furthermore, Δpep27ΔcomD immunization significantly increased the survival time after heterologous challenges, and diminished colonization levels independent of serotype [28], as shown in Figure 2 and Table 1.

Mechanistically, *vncRS* is activated by lactoferrin in serum and is required for the development of pneumonia and sepsis. When the VncS sensor is exposed to lactoferrin, it is phosphorylated, and the phosphate group is transferred to the VncR response regulator, allowing the VncRS operon to be induced, which in turn secretes the effector Pep27, which is thought to cause bacterial lysis and release, leading to host lung inflammation. Deletion of the effector Pep27 does not induce lysis, and it is incompetent for invasion into the lungs and blood, confirming that *pep27* is essential for the inflammatory response and sepsis [24].

## 3. Gut–Brain, Gut–Lung, and Gut–Liver Axis

The gut contains a variety of substances, including food, medications, if any, and secretions from the body, such as stomach acid and bile salts. As such, our gut microbiome can be altered by these substances. While there are many types of microorganisms in the gut, what they are fed can select for certain microorganisms, which in turn produces specific microbial products, and/or metabolites. In the gut, various hydrolyzed and transformed products and metabolites, such as short-chain fatty acids (SCFAs), dopamine, tyrosine, tryptophan, trimethylamine N-oxide, and urolithin A, are present and involved in gut–brain axis (GBA) communication. These metabolites are absorbed through the intestinal wall and enter the bloodstream, where they are eventually distributed to all organs. However, some of the metabolites enter through the blood–brain barrier and affect or modulate the central nervous system [36]. However, the GBA network appears to be interconnected by the vagus nerve, with signals from the gut reaching the afferent vagus nerve in the brain; in return, the CNS sends signals to the gut via efferent vagal neurons. Thus, the gut microenvironment can be monitored via gut neurons and transmitted to the brain via the vagus nerve, and vice versa [37]. The GBA plays an important role in homeostasis to regulate various functions of the gut and brain, including immune function, barrier permeability, and the gut reflex [37,38,39,40].

A variety of microbial metabolites can exert different effects; for example, butyric acid mitigates cognitive impairment and taurine enhances memory [36]. Thus, the gut microbiome can actually act as a multifactorial cause of gut and brain diseases [38,39,41]. Increased populations of harmful bacteria and abnormal gut microbiota composition can lead to increased gut dysbiosis and immune dysfunction through GBA interactions, which can lead to brain disorders such as depression, high sensitivity to stress, and neurodegenerative disorders [37,38,39,41]. The gut microbiome can modulate brain structure and function, and in turn, the brain can modulate the microbial environment and microbiome composition within the gut microbiome. Collectively, these pathways are referred to as the microbiota–gut–brain axis (microbiota–GBA), which represents a comprehensive concept of biochemical signaling and interactions between the brain, gut bacteria, and gastrointestinal tract [38].

Accumulating evidence suggests that SCFAs consist of acetate, lactate, butyrate, propionate, and succinate, which are produced through fermentation of fiber-rich diet by bacteria (e.g., *Prevotella*, *Bacteroides*, and *Ruminococcus*) [42]. High-fructose diets reduce SCFAs, promote gut dysbiosis, and induce neuroinflammation [43]. SCFAs can be considered a leading link in the immune axis between the gut and lungs. Increased luminal butyrate production promotes mucosal healing and encourages the production of protective mucus along the intestinal epithelium [44]. Indeed, SCFAs are known to modulate immune homeosis and mucosal defense, thus contributing to barrier functions. Several *Lactobacillus* species are known to secrete lactate-producing bacteria, a precursor for SCFA-producing bacteria.

Interestingly, alterations in the nasal microbial community including airways also affect the composition of intestinal microbiota. In addition, constant exposure of mucosal surfaces, particularly the respiratory and gastrointestinal tracts, to microbes and antigens from the environment makes these surfaces valuable for shaping tolerogenic responses in autoimmune and allergic disease. Numerous studies have shown that 2·5 μL of inoculum consisting of fluids, particles, or even microorganisms deposited into the nasal cavity of mice can later be detected in the gastrointestinal tract (GIT) [45]. This indicates that the mucosal immune system of the GIT may serve as a primary sensor of any foreign antigens that are introduced into the nasal cavity [46]. For example, manifestations of pneumonia due to *Pseudomonas aeruginosa* or multi-drug resistant *Staphylococcus aureus* in lungs are believed to trigger gut injury [47]. Furthermore, several gastrointestinal disorders have manifestations in the respiratory tract, for example, about half of IBD patients with known alterations in their intestinal microbiota composition have abnormal lung function. COPD (chronic obstructive pulmonary disease) patients show intestinal hyper-permeability with a high prevalence of IBD [48], thus suggesting that the “gut–lung axis” is a bi-directional communication network where many respiratory infections are often accompanied by gastrointestinal symptoms [49]. Communication in the gut–lung axis comprises many direct and indirect pathways. Moreover, disturbing the lung microbiome with the antibiotic neomycin can significantly attenuate the severity of the experimental autoimmune encephalomyelitis (EAE) by sensitizing the autoreactivity of brain T cells via microglia. Thus, lung microbiome dysbiosis can regulate brain autoimmunity [50]. Although the number of bacteria in the lungs is much lower than that in the gut, it has been shown that changes in the lung microbiome could affect the brain, perhaps through the lung–brain axis, which is functionally similar to the gut–brain axis.

Another organ involved in the regulation of the gut microbiome is the liver, which also affects gut function through the gut–liver axis. Bile salts, which are necessary for lipid digestion in vertebrates, are synthesized in the liver, conjugated with amino acids such as glycine and taurine by primary host enzymes, and secreted into the small intestine. Once secreted, bile salts are deconjugated by secondary gut bacteria that possess bile salt hydrolase (BSH) to separate amino acids from bile salts [51]. A variety of gut bacteria can conjugate bile acids, including the genera *Actinobacterium*, *Bacillus*, *Bacteroides*, *Bifidobacterium*, *Clostridium*, *Fusobacterium*, and *Lactobacillus*. Furthermore, bile amidates have also been produced by the genera *Clostridium*, *Bifidobacterium*, and *Enterococcus* [52]. In addition, deconjugation of glycine-conjugated bile acids is associated with the genus *Gemmiger*, and deconjugation of taurocholic acid is linked to the genera *Eubacterium* and *Ruminococcus* [39]. This suggests that different microbes may have the same substrate specificity and may share functions with other microbes. Moreover, gut microbiota imbalances and changes in bile acids are involved in the regulation of inflammatory responses through bile acids receptors, but conversely, bile acid receptor changes can also affect the abundance of gut microbes [53].

The bile salt hydrolase (*bsh*) gene is characterized as an acyltransferase that forms bile acid amidates [53,54,55]. Furthermore, human gut metagenomic analysis has shown that BSH activity and abundance in the human gut is associated with IBD [56]. Bacterial bile acid amidates act as ligands and activate host receptors responsible for the transcription of the aryl hydrocarbon receptor (AHR) and the Pregnane X receptor (PXR) [55]. PXR levels correlate well with intracellular bile acid levels in the gut and liver, and inhibiting PXR increases IL-8 and TNF-α and decreases IL-10 and TGF-β, suggesting a worsening of IBD symptoms. Thus, PXR agonist exhibited anti-inflammatory effects by inhibiting NF-κB and suppressing cytokine secretion in mice with dextran sulfate sodium (DSS)-induced colitis [53].

Elevated levels of gut bacteria-derived secondary bile acids (i.e., lithocholic acid (LCA) and deoxycholic acid (DCA)) harm the intestinal barrier, leading to dysbiosis and increased gut inflammatory responses. In addition, a lack of BSH-positive bacteria can lead to intestinal inflammation and bile acid metabolic dysfunction [57]. Increasing intestinal BSH activity by administering BSH-competent probiotics or introducing them via fecal transplantation can provide various health benefits to the host. Therefore, to overcome gut inflammation, research exploring the regulation of the farnesoid X receptor (FXR, including PXR), the master regulator of bile acid homeostasis, and its impact on the gut microbiome is of considerable importance [57].

Since bile acids can be converted into different metabolites by various gut bacteria, it is necessary to determine which bacterial enzymes degrade/transform bile acids and how they affect the gut. Also, humans and rodents differ significantly in terms of the regulatory activity of certain bile acids on FXR receptors, so rodent results should be considered with caution when comparing them to human results. To fully understand bile acid-dependent FXR function, especially in mucosal immune regulation, we need to understand how different parameters in rodents and humans change in response to inflammation, diet, and microbial imbalance [53].

## 4. Treg Cells for Brain Diseases

Impaired Treg function can be modulated by either functional deficiency including impaired stability of Treg cells or numerical deficiency of Treg cells. Impaired stability of Treg cells seems to be the cause of autoimmune diseases such as multiple sclerosis (MS). To overcome this conundrum, several methods have been developed [58]. To reverse low Treg stability prior to the implementation of Treg therapy, rapamycin (mTORC1 inhibitor) could restore Treg cells in autoimmune disease [59].

Immunologically, the brain acts as a part of the systemic immune response system, connecting not only the gut–brain axis but also the neural network, and Tregs play a pivotal role in inflammation. In particular, an emerging view is that brain Treg cells directly support tissue regeneration and repair processes by suppressing glial reactivity to neuronal damage. Brain Treg cells are attractive therapeutic targets in all neurological diseases, from neuroinflammatory disorders to neurodegenerative diseases and even psychiatric disorders [60].

The GBA has a bi-directional association. Thus, brain function is affected by intestinal inflammation, and conversely, brain disorders can cause IBD. Similarly, Treg function is controlled by GBA in a bi-directional manner via the neuroimmune response. SCFAs are known to limit mucosal inflammation via the induction of Tregs [61]. Consistent with this, intranasal immunization of Δpep27 showed that the microbiome composition of Δpep27-immunizaed colitis mice was positively correlated with gut Treg induction and negatively associated with proinflammatory cytokines [31], presumably via the GBA or the lung–gut axis.

One of the triggering mechanisms of Treg that act in this bi-directional way is mediated by liver vagal afferent nerves, which indirectly sense the gut environment and relay the sensing inputs to the brain parasympathetic nerves as well as enteric neurons. This therefore shows that brain Treg upregulation also upregulates gut Treg function via the vagus nerve [40]. Thus, intestinal homeostasis may be regulated by the hepatic vago-vagal GBA reflex to maintain intestinal Treg cell numbers [40], suggesting that treating intestinal inflammation using brain Treg upregulation may be a more feasible therapeutic approach than modulating intestinal Treg function by administering other chemicals or neuromodulators in the future. Studies are underway to enhance brain Treg function to treat multiple sclerosis [62,63], Parkinson’s disease [64], glioblastoma [65], and other neurological diseases such as epilepsy, neurotrophic pain, and stroke [58,66,67]. Tregs can be classified as either natural Tregs (nTregs) or induced Tregs (iTregs). Enhancement of nTregs can be achieved by (1) stabilizing nTreg function and survival, (2) manipulating antigen-specific Treg reactions, (3) using immunomodulators peripherally to induce Treg populations, or directing the adoptive transfer of Treg cells. iTreg populations are generated via direct administration of low doses of Treg inducers such as IL-2, CD3 monoclonal antibody, or GM-CSF. In addition, the use of these immunologic agents to proliferate dysfunctional Tregs ex vivo and then perform autologous adoptive transplantation is being investigated [58]. Several approaches that combine these therapies with brain delivery methods to enhance efficacy are also currently under intensive development [60].

## 5. Treg Cells for Inflammatory or Allergic Diseases

Currently, novel microbiome approaches to overcome IBD include either fecal microbiome transplantation (FMT) [68,69,70] or ingestion of microbial strains that induce Treg function that can suppress intestinal inflammation [71,72]. However, these methods require antibiotic treatment to eliminate microbial imbalances prior to bacterial treatments. Vancomycin is administered to purge vegetative *C. difficile*, which produces toxins and causes inflammation and diarrhea, but does not kill the spore forms that cause germination when treatment is discontinued. This is because antibiotic treatment causes a lack of beneficial Firmicutes and results in an increase in bile acid, which in turn allows germination of *C. difficile* spores. Therefore, discontinuation of antibiotic therapy and/or incomplete sterilization may cause a recurrence of *C. difficile* disease. Recently, oral administration of SER-109 (from the fecal microbiota of healthy donors) has been shown to significantly reduce the recurrence rate of *C. difficile* (12%) compared with that of the placebo group (40%) [69]. Moreover, SER-109 resulted in a significant improvement in disease-specific quality of life scores from as early as week 1 compared with those of patients treated with the placebo, with steady and sustained improvement continued through to week 8 post-dose [70]. Moreover, the gut microbiome is subject to dietary modifications even after these treatments [73,74]. Therefore, an alternative approach that is not affected by diet would be preferable, and more efficient methods of Treg induction and maintenance are required. To date, it is unknown whether nasal immunization can upregulate Tregs in the skin and gut due to the lack of characterization or vaccination of the nasal mucosa.

Although intestinal Treg enrichment has been studied, the intestinal Treg population is continuously modulated by diet and other medications and interacts reciprocally with the gut–brain, gut–lung and gut–liver networks to maintain homeostasis [75,76,77]. Diets and drugs targeting specific microbiomes are available to improve host disease, but most do not seem to be universally applicable [78]. Therefore, even if Treg cells are enriched with a specific diet, medication, or appropriate therapy, the Treg population remains intact while under the control of diet and medication. Therefore, one of the key factors in actually achieving a Treg-enhancing effect is Treg cell stability and/or Treg resilience.

## 6. Intranasal Immunization of Δpep27 Protects against Pathogens and Influenza Virus Infection

In our approach to intranasal immunization using Δpep27 for the prevention of pneumococcal diseases, microarray and system biology analyses of human lung cells after Δpep27 infection unraveled unexpected features predicting the preventive effect of influenza virus infection and intestinal abnormality. Thereafter, a series of experiments on this prediction were performed and showed that the prediction was true [26,31,32]. Δpep27 provides transient non-specific protection from heterologous bacteria through non-canonical Wnt upregulation. Nasal immunization with Δpep27 can inhibit colonization by *Staphylococcus aureus* and *Klebsiella pneumoniae*, indicating non-specific resistance to respiratory pathogens [29].

Injectable pneumococcal vaccines, including the 23-valent polysaccharide vaccine and the 13-valent conjugate vaccine, do not provide mucosal immunity and do not provide complete protection against secondary pneumococcal infection following primary influenza virus infection [79]. To address these challenges, we determined whether or not Δpep27 could protect mice against secondary pneumococcal infection following influenza virus infection. Surprisingly, Δpep27 protected mice against secondary pneumococcal infection after influenza virus infection by lowering the influenza virus burden in the lungs. In contrast, the unimmunized group of mice had a nearly 60% higher mortality rate following pneumococcal infection due to higher bacterial loads. Δpep27 vaccination alone can prevent influenza and pneumococcal infections by reducing viral titers in the lungs after infection. Overall, Δpep27 immunization is a novel and safe method to overcome both invasive pneumococcal disease and serious secondary infections following influenza infection during influenza epidemics [26], as shown in Figure 2 and Table 1.

During pneumococcal pneumonia, phagocytes produce H_2_O_2_ and reactive oxygen species (ROS) for bacterial removal; nonetheless, the lung is vulnerable to these oxidative stresses, resulting in extensive cellular and lung damage [80]. Thus, we investigated the therapeutic effect of ∆pep27 immunization on antioxidant small proline-rich repeat (SPRR) genes in the lungs and its associated consequences on the gut dysbiosis. We observed that Δpep27 significantly increased the levels of the SPRR genes in the lungs, suggesting a strengthened alveolar barrier and enhanced resistance to external stressors, resulting in a robust regenerative and oxidant-stress-relieving mechanism to re-establish immunological tolerance [32,81]. Additionally, SPRR genes are involved not only in the establishment of the physical barrier but also in cell migration and wound healing [81,82]. Δpep27, on the other hand, significantly increased the expression of *SPRR* genes, resulting in a more strengthened alveolar barrier and enhanced resistance to external including pneumococci [32,81]. Similarly, Korean Red Ginseng (KRG), a traditional medicinal herb widely used as an immune booster, enhanced the efficacy of the Δpep27 vaccine by increasing the survival rate of mice infected with pneumococcus by inhibiting ROS production, suppressing ERK signaling-mediated cell death and reducing inflammation [30]. This suggests that Δpep27 immunization blocks ROS and oxidative stress [30,32], as shown in Figure 2 and Table 1.

Macrophages are classified into M1 and M2 macrophages, which produce inflammatory and anti-inflammatory cytokines, respectively [83]. Adoptive transfer of M2 macrophages or the induction of M2 polarization has been shown to suppress experimental colitis [84]. M2 macrophages aid in the resolution of inflammation by downregulating inflammatory cytokines and secrete copious amounts of IL-10 and TGF-β, thereby protecting against colitis to promote tissue repair and driving epithelial cell regeneration [83]. Intranasal Δpep27 immunization upregulates colonic M2 macrophages, thereby inhibiting inflammatory milieu [32]. Tregs maintain homeostasis by suppressing excessive immune activation. Tregs are also very well defined as helpful for resolving and repairing lung damage caused by infection [85].

Neutrophils play an important role in eliminating pathogens during respiratory infections, and when they are mobilized to the lungs where pathogens have invaded, they phagocytose the pathogens and then digest and kill them by producing reactive oxygen species. However, in bacterial pneumonia, this process can lead to excessive lung damage and respiratory failure. Therefore, neutrophil phagocytosis (efferocytosis), the phagocytic removal of dead or dying neutrophils, is a key process in resolving lung inflammation. Tregs interact with alveolar macrophages to promote neutrophil phagocytosis and promote recovery. Overall, Tregs resolve inflammation and orchestrate tissue protection and airway system repair in mice and humans [85,86,87,88]. After three intranasal vaccinations of Δpep27, fluorescence-activated cell sorting (FACS) analysis of splenocytes showed an increase in Treg expression proportional to the number of immunizations. Additionally, it was confirmed that Tregs were induced in serum and bronchoalveolar lavage fluid (BALF) (Kim GL, manuscript in preparation). Thus, Tregs may be one of the mechanisms by which pathogen infection is defended against by intranasal Δpep27 immunization.

## 7. Intranasal Immunization of Δpep27 Protects Allergic Diseases

Inactivated serotype 3 *S. pneumoniae* has been reported to be effective against allergic diseases, including asthma, via Treg upregulation. These inactivated strains in a mouse model significantly suppressed the allergic inflammatory responses that are pivotal in the development and progression of asthma, including Th1 and Th2 cytokine production and eosinophil recruitment to the airways during or after ovalbumin sensitization [89,90], but they are toxic and cannot be used as a vaccine.

Intranasal Δpep27 immunization before or after allergen exposure could restore the necessary balance of Th1/Th2 cells by reducing Th2 activity and maintaining Th1 and Treg activity disturbed during asthma. Additionally, allergic airway inflammation in the lung was significantly reduced by Δpep27 immunization. Δpep27 immunization may provide long-term protection against asthma without any toxicity [33].

In addition, Δpep27 immunization alleviated allergic symptoms such as sneezing and rubbing frequency and reduced TLR2 and TLR4 expression, Th2 cytokines, and eosinophil infiltration in the nasal mucosa of an ovalbumin (OVA)-induced allergic rhinitis mouse model [34], shown in Figure 2. Mechanistically, Δpep27 reduced the activation of the NLRP3 inflammasome in the nasal mucosa by downregulating the TLR signaling pathway and subsequently prevented allergic reactions [34].

In asthma models, IL-27 is an anti-inflammatory cytokine that belongs to the IL-12 family and is primarily expressed on dendritic cells, macrophages, and monocytes [91]. Because IL-27 targets Tregs in asthma models [92] and reduces respiratory allergy symptoms [93,94], the asthma-relieving effects of Δpep27 appear to be due to the suppression of inflammation by IL-27-induced Tregs.

In bronchial asthma, inflammation is frequently triggered repeatedly, resulting in increased microvascular permeability and edema, which eventually leads to airway remodeling and a thickening of the airway walls [95]. Injection of Treg cells after the onset of chronic asthma disease prevented airway remodeling [96], indicating that Tregs can resolve chronic inflammation in asthma models. Thus, Tregs induced by Δpep27 may also contribute to resolving airway remodeling. In contrast to Tregs, IL-27 appears to act differently in asthma suppression; when IL 27 was administrated therapeutically, IL 27 did not ameliorate airway inflammation, airway hyperresponsiveness, and airway remolding. Prophylactically, administration of IL-27 decreased the concentration of Th2 cytokines and increased the number of type 1 regulatory T (Tr1) cells in the lungs [97,98], suggesting that IL-27 may prevent asthma. Thus, they demonstrated that IL-27 may be useful for preventive purposes but not for therapeutic purposes. pep27, however, works as both a therapeutic and preventive method.

Prophylactic and therapeutic analysis showed that Δ*pep27* could elicit anti-inflammatory Treg-relevant factors and epithelial barrier genes (filaggrin, involucrin, loricrin, and SPRR proteins). Accordingly, pneumococcal Δpep27 immunization upregulated Treg activity, suppressing epidermal collapse, IgE, and Thymic stromal lymphopoietin (TSLP). On the other hand, Treg suppression worsened atopic dermatitis through the upregulation of TSLP and Th2 and the repression of epithelial barrier function compared with the non-suppressed pneumococcal Δpep27 group. In summary, pneumococcal Δ*pep27* immunization alleviated allergic dermatitis symptoms by upregulating Tregs and epithelial barrier functions and suppressing TSLP and Th2 to relieve allergic dermatitis symptoms [35].

## 8. Intranasal Immunization of Δpep27 Potentially Protects IBD via Anti-Oxidative SPRR and Anti-Inflammatory M2 Upregulation through Treg Induction

Intranasal Δpep27 immunization prevented DSS-induced colitis. ∆pep27 significantly mitigated oxidative stress parameters and downregulated pro-inflammatory cytokines and Wnt5a expressions via Treg induction in the gut. Moreover, ∆pep27 induces upregulation of the anti-inflammatory genes IL-10 and TGF-β1, as well as M2 macrophages, via Treg induction and tight junction genes. Δpep27 also suppresses DSS-induced caspase-14 expression and upregulates Tregs, resulting in healthy microbiota. Inhibition of Treg function confirmed that ∆pep27 has therapeutic effects on gut inflammation and caspase-14 via Treg upregulation. Overall, intranasal immunization with ∆pep27 can attenuate colonic inflammation via Treg induction and could be a highly pragmatic way to re-establish immunological tolerance [31,32], as shown in Figure 2 and Table 1.

## 9. Conclusions

To date, treating allergic diseases as well as recurrent inflammatory diseases, including infections and IBD, remains a challenging task. Despite representing a complex approach that evaluates immunogenicity, mucosal vaccines have the potential to improve current applications and extend the utility of vaccines for diseases such as IBD and allergies. Although there have been many attempts to use Tregs, which are effective for hypersensitivity reactions such as excessive inflammation or allergies, there are many limitations in terms of functionality. We confirmed that Tregs induced via intranasal immunization were consistently expressed not only in the nasopharynx and lungs, but also in the skin and intestines, rendering them effective against inflammation/hypersensitivity reactions. Mechanistically, Δpep27 suppresses oxidative stress levels, which are closely linked to gut dysbiosis, potentially by increasing the SPRR family in the lungs, suggesting that the lung–gut axis is a bi-directional communication network. Furthermore, analysis of key genes in the lungs induced by Δpep27 immunization highlighted mucosal protection, particularly in the lungs and gut, emphasizing the role of Tregs in establishing immune tolerance. Furthermore, Δpep27 immunization induced M2 macrophages, an antioxidant milieu, to mitigate the stress response, and Treg attenuated caspase-14 and Wnt5a expression independent of the inflammatory environment through the lung–gut axis, suggesting robust anti-inflammatory mucosal tolerance and subsequent restoration of the gut microbiota, ensuring that barrier integrity is maintained to ensure intestinal immune homeostasis. ∆pep27 immunization appears to promote the development and restoration of functional Treg cells in the skin, and internal and respiratory organs, presumably through mucosal Treg infiltration or induction, as well as on the gut–liver and gut–brain axis, which in turn may cooperate with the brain or liver to exert anti-inflammatory effects on the skin and other organs. Moreover, the lung microbiome can modulate autoimmune diseases in the brain, indicating that the lung–brain axis may functionally operate similarly to the gut–brain axis (Figure 2, Table 1). To this end, future studies are warranted to determine whether the pep27 vaccine also induces Tregs in the brain and liver. Overall, ∆pep27 may be a promising mucosal vaccine candidate therapy for clinical applications in allergic and inflammatory diseases (Figure 1 and Figure 2, Table 1). However, due to the complicated nature of IBD and allergic diseases, further clinical trials optimizing the parameters regulated by Δpep27 and the resilience of food-induced changes in Treg expression are needed to confirm its effectiveness in these diseases.

## Figures and Tables

**Figure 1 vaccines-12-00737-f001:**
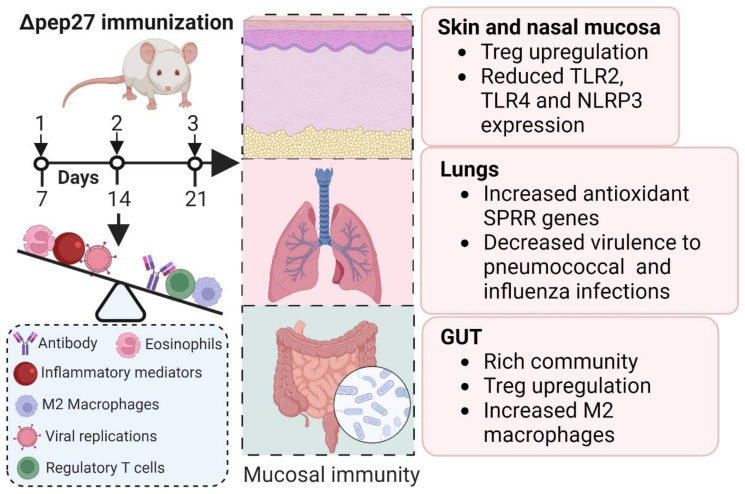
∆pep27 immunization in various inflammatory diseases. Mice were intranasally immunized with ∆pep27, 3 times, once a week, with 1 × 10^8^ CFU. Experimental diseases were induced via the appropriate methods for specified days until the mice were sacrificed. Collectively, several parameters such as IgG level, M2 macrophages, and Treg cells were induced via ∆pep27 immunization. However, inflammatory mediators using cytokine and chemokine transcription factors in pneumococcal and virus models were suppressed. Image created using the BioRender application (biorender.com).

**Figure 2 vaccines-12-00737-f002:**
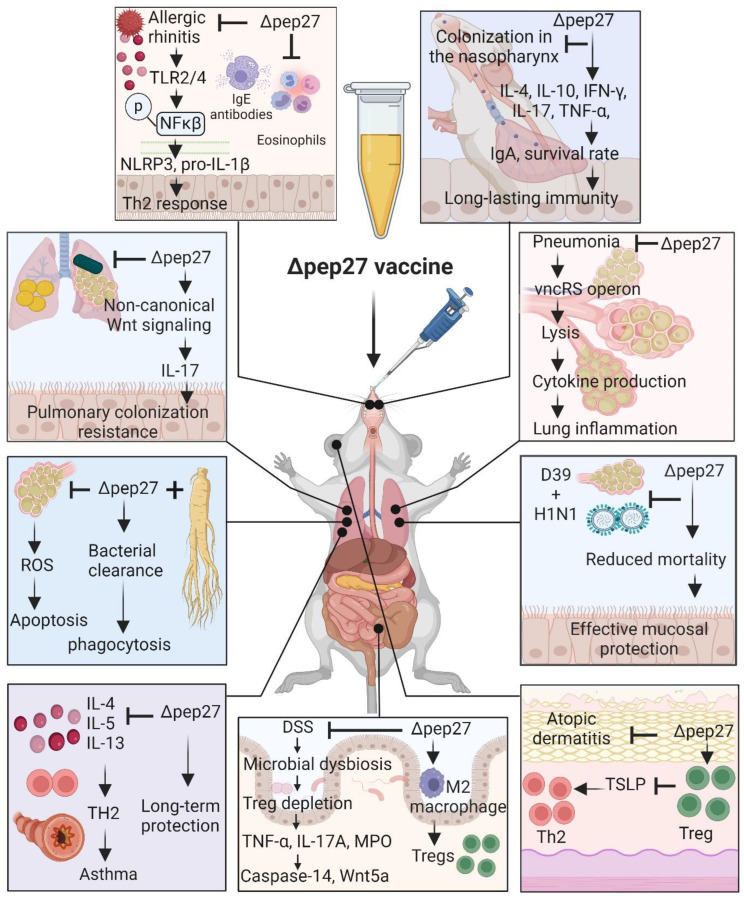
Therapeutic effect of pep27 mutant vaccine in various inflammatory diseases. The response to Δpep27 immunization via the intranasal route was analyzed using gene signatures predominant in different organs of mice. Δpep27 immunization attenuated disease development in the mouse model by suppressing aberrant gene expression and dysregulated immune responses. These disturbances deplete Treg cells and subsequently induce inflammatory mediators, including cytokines, chemokines, and inflammatory pathways. In contrast, ∆pep27 had a remedial effect that preserved mucosal integrity through Treg upregulation, suggesting a promising candidate therapy for clinical application with a potent anti-inflammatory mucosal immune mechanism. Image created using the BioRender application.

**Table 1 vaccines-12-00737-t001:** Effects of Δpep27 on pneumococcal virulence and Δpep27 immunization efficacy in multiple disease models.

Disease Model	Δpep27 Mechanism	Ref.
Lethal pneumococcal D39 strain	Decreases virulence and allows rapid clearance due to lower levels of capsular polysaccharide	[18]
Lethal pneumococcal pneumonia	Δpep27-attenuated lactoferrin induced the *vncRS* operon to prevent lysis, in vivo cytokine production, and subsequent lung inflammation	[24]
Lethal pneumococcal pneumonia	Shows resistance to lysis and reduced cytotoxicity, resulting in decreased inflammation and death, enabling effective mucosal protection	[25]
Lethal pneumococcal pneumonia	Reduces morbidity and mortality against pneumococcal and influenza infections	[26]
Pneumococcal pneumonia	Δpep27 immunization impairs serotype-independent colonization by increasing IgA-, Th1-, and Th17-type cytokine responses	[27]
Pneumococcal colonization	Non-transformable Δpep27ΔcomD immunization significantly diminished colonization levels regardless of serotype	[28]
*Nasal infection* with *S. pneumoniae*, *S. aureus*, *K. pneumoniae*	Δpep27 inhibited colony formation of pathogens and induced noncanonical Wnt and subsequent IL-17 secretion	[29]
Korean Red Ginseng + Δpep27 before lethal pneumococcal challenge	Korean Red Ginseng enhanced Δpep27 vaccine efficacy by inhibiting ROS production, apoptotic signaling, and inflammation	[30]
DSS-induced Colitis	Δpep27 significantly attenuated the expression levels of pro-inflammatory cytokine caspase-14 to attenuate experimental colitis through the restoration of functional Tregs and healthy gut microbiota composition	[31]
DSS-induced Colitis	Tregs elicited by Δpep27 were able to suppress Wnt5a expression to help restore immunological tolerance and provide a robust antioxidant milieu	[32]
Ovalbumin-induced asthma	Δpep27 immunization suppresses TH2 cytokine and pulmonary eosinophil accumulation, and goblet cell proliferation, maintaining a balance between Th1, Th2, and Treg cells	[33]
Ovalbumin-induced Allergic rhinitis	Δpep27 reduced the activation of the NLRP3 inflammasome in the nasal mucosa by suppressing NF-κB activation through downregulating TLR2 and TLR4 expression	[34]
Oxazolone-induced Atopic dermatitis	Δpep27 upregulated Treg and epithelial barrier function and inhibited TSLP and Th2 expression	[35]

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
