# Peer review of "Intranasal Immunization of Pneumococcal pep27 Mutant Attenuates Allergic and Inflammatory Diseases by Upregulating Skin and Mucosal Tregs"

_vaccines, 2024, doi:10.3390/vaccines12070737_

Round 1

Reviewer 1 Report

Comments and Suggestions for Authors

Hamid Iqbal and Dong-Kwon Rhee, in this review, discussed the implications of vaccination with the pneumococcal pep27 mutant in treating and preventing allergic and inflammatory diseases. They discussed the impact of the microbiome on immunity, the role of Treg cells in controlling inflammation, and how Δpep27 immunization-induced Tregs can protect against allergic and other inflammatory diseases. The relevant studies are thoroughly summarized and discussed. However, there are a few points that need to be addressed to improve this review:

1. Section 6: “Intranasal immunization of Δpep27 protects against pathogens and influenza virus infection”: It is indicated that this protection is mediated through non-canonical Wnt upregulation and inhibition of pathogen colonization. The role of Treg cells is unclear here. In fact, Tregs may suppress the immune response needed to kill pathogens. The authors should ensure this section aligns well with the central theme and title of the review.

2. Line 11: It lacks clarity and context regarding which therapy leads to adverse events.

3. Line 50: This should be rephrased. Just because one case of adjuvanted nasal vaccine caused Bell’s palsy, it is not justified to conclude, “Therefore, nasal immunization should be administered without adjuvants.” It should caution the careful selection of adjuvants in nasal vaccine design instead.

4. Section “2. Gut-brain, gut-lung, and gut-liver axis”: This section is overly discussed. The paragraph should focus on how a certain microbe (particularly the pneumococcal pep27 mutant) impacts the immune system (specifically the induction of Treg cells). It is acceptable to include a general discussion on this topic; however, it should be concise and more focused.

Author Response

Hamid Iqbal and Dong-Kwon Rhee, in this review, discussed the implications of vaccination with the pneumococcal pep27 mutant in treating and preventing allergic and inflammatory diseases. They discussed the impact of the microbiome on immunity, the role of Treg cells in controlling inflammation, and how Δpep27 immunization-induced Tregs can protect against allergic and other inflammatory diseases. The relevant studies are thoroughly summarized and discussed. However, there are a few points that need to be addressed to improve this review:

  1. Section 6: “Intranasal immunization of Δpep27 protects against pathogens and influenza virus infection”: It is indicated that this protection is mediated through non-canonical Wnt upregulation and inhibition of pathogen colonization. The role of Treg cells is unclear here. In fact, Tregs may suppress the immune response needed to kill pathogens. The authors should ensure this section aligns well with the central theme and title of the review.

Ans. Authors appreciate the referees for understanding and acknowledging the impact of our review by providing us with valuable suggestions to undoubtedly enhance the credibility of our manuscript. We hope that our explanation to address the referee’s concerns will satisfy the reviewer in the revised manuscript.

To address the comments, we described how Tregs function during pathogenic infections, particularly pneumococcal infections, and presented unpublished work to show why Tregs can defend against acute inflammation and increase recovery from pneumococcal and viral infections as follows;

Therefore, in the revision, we’ve added the following to line 363.

“Tregs maintain homeostasis by suppressing excessive immune activation. Tregs are also very well defined to help resolve and repair lung damage caused by infection [Jovisic M, et al., 2023]. Neutrophils play an important role in eliminating pathogens during respiratory infections, and when they are mobilized to the lungs where pathogens have invaded, they phagocytose the pathogens and then digest and kill them by producing reactive oxygen species. However, in bacterial pneumonia, this process can lead to excessive lung damage and respiratory failure. Therefore, neutrophil phagocytosis (efferocytosis), the phagocytic removal of dead or dying neutrophils, is a key process in resolving lung inflammation. Tregs interact with alveolar macrophages to promote neutrophil phagocytosis and promote recovery. Overall, Tregs resolve inflammation and orchestrate tissue protection and airway system repair in mice and humans [D'Alessio FR et al, 2009; Jovisic M et al., 2023; Proto JD et al, 2018; Xu R et al, 2023]. After three intranasal vaccinations of Δpep27, fluorescence-activated cell sorting (FACS) analysis of splenocytes showed an increase in Treg expression proportional to the number of immunizations. Additionally, it was confirmed that Tregs were induced in serum and bronchoalveolar lavage fluid (BALF) [Kim GL, Manuscript in preparation]. Thus, Tregs may be one of the mechanisms by which pathogen infection is defended against by intranasal Δpep27 immunization.”

  1. Line 11: It lacks clarity and context regarding which therapy leads to adverse events.

Ans. We changed the word from current to conventional.

  1. Line 50: This should be rephrased. Just because one case of adjuvanted nasal vaccine caused Bell’s palsy, it is not justified to conclude, “Therefore, nasal immunization should be administered without adjuvants.” It should caution the careful selection of adjuvants in nasal vaccine design instead.

Ans. Thank you for your suggestion. We changed it according as ‘Therefore, nasal immunization is preferred to administer without adjuvants to avoid pathological conditions’.

  1. Section “2. Gut-brain, gut-lung, and gut-liver axis”: This section is overly discussed. The paragraph should focus on how a certain microbe (particularly the pneumococcal pep27 mutant) impacts the immune system (specifically the induction of Treg cells). It is acceptable to include a general discussion on this topic; however, it should be concise and more focused.

Ans. We agree with the reviewer comment and intended to highlight the significance of Gut-brain, gut-lung, and gut-liver axis due to following reasons.

  1. As mentioned, pep27 gene encodes an autolysis-inducing factor in pneumococcal strain (type 2, D39) possess increase bacterial burden and colonization in the nasopharynx and is thus responsible to mediate virulence factor in the lung [Kwon MK, 2008].
  2. However, inactivation of pep27 gene (Δpep27) makes the pneumococci non-lytic [Lee S et al., 2018], and incapable of invading into the lungs, blood, and brain [Kim EH et al., 2012], resulting in a virtually non-cytotoxic and highly safe agent that did not cause death after injection into the brains of immunocompromised mice [Lee S et al., 2018]. Furthermore, Δpep27 significantly attenuated experimental colitis through restoration of functional Tregs and healthy gut microbiota composition [Iqbal et al, 2022].
  3. The purpose of Gut-brain, gut-lung, and gut-liver axis to discuss was to configure the correlation of genes using system biology analysis regulating the mucosal surfaces by ∆pep27 immunization in the nasopharynx and its predominant effect in the lungs, gut and brain to establish immune tolerance mechanism.

  1. Reviewer also pointed towards certain microbes (particularly the pneumococcal pep27 mutant) impacts the immune system (specifically the induction of Treg cells).

  1. We already added the required information in Inflammatory bowel disease article [Iqbal et al, 2022], where our findings indicated that microbiota composition in Δpep27-DSS mice showed positive correlations with the Treg induction and negative association with the proinflammatory cytokines interaction revealing a major role for the microbiota in shaping the repertoire, number, and activation of Tregs.

 References

D'Alessio, F.R.; Tsushima, K.; Aggarwal, N.R.; West, E.E.; Willett, M.H.; Britos, M.F.; Pipeling, M.R.; Brower, R.G.; Tuder, R.M.; McDyer, J.F.; King, L.S. CD4+CD25+Foxp3+ Tregs resolve experimental lung injury in mice and are present in humans with acute lung injury. J Clin Invest. 2009,119, 2898-913. doi: 10.1172/JCI36498.

Iqbal, H.; Kim, G.L.; Kim, J.H.; Ghosh, P.; Shah, M.; Lee, W.; Rhee, D.K. Pep27 Mutant Immunization Inhibits Caspase-14 Expression to Alleviate Inflammatory Bowel Disease via Treg Upregulation. Microorganisms. 2022a, 10, 1871. doi: 10.3390/microorganisms10091871.

Jovisic, M.; Mambetsariev, N.; Singer, B.D.; Morales-Nebreda, L. Differential roles of regulatory T cells in acute respiratory infections. J Clin Invest. 2023, 133, e170505. doi: 10.1172/JCI170505.

Kim, E.H.; Choi, S.Y.; Kwon, M.K.; Tran, T.D.; Park, S.S.; Lee, K.J.; Bae, S.M.; Briles, D.E.; Rhee, D.K. Streptococcus pneumoniae pep27 mutant as a live vaccine for serotype-independent protection in mice. Vaccine. 2012, 30, 2008-19. doi: 10.1016/j.vaccine.2011.11.073.

Kwon, M.K. Mutagenesis of the pneumococcal genes induced during infection into the lung cells and characterization of the mutants in virulence. MS Thesis, School of Pharmacy, Sungkyunkwan University. 2008.

Lee, S.; Ghosh, P.; Kwon, H.; Park, S.S.; Kim, G.L.; Choi, S.Y.; Kim, E.H.; Tran, T.D.; Seon, S.H.; Le, N.T.; Iqbal, H.; Lee, S.; Pyo, S.; Rhee, D.K. Induction of the pneumococcal vncRS operon by lactoferrin is essential for pneumonia. Virulence. 2018, 9, 1562-1575. doi: 10.1080/21505594.2018.1526529.

Proto, J.D.; Doran, A.C.; Gusarova, G.; Yurdagul, A. Jr.; Sozen, E.; Subramanian, M.; Islam, M.N.; Rymond, C.C.; Du, J.; Hook, J.; Kuriakose, G.; Bhattacharya, J.; Tabas I. Regulatory T Cells Promote Macrophage Efferocytosis during Inflammation Resolution. Immunity. 2018, 49, 666-677.e6. doi: 10.1016/j.immuni.2018.07.015.

Xu, R.; Jacques, L.C.; Khandaker, S.; Beentjes, D.; Leon-Rios, M.; Wei, X.; French, N.; Neill, D.R.; Kadioglu, A. TNFR2+ regulatory T cells protect against bacteremic pneumococcal pneumonia by suppressing IL-17A-producing γδ T cells in the lung. Cell Rep. 2023, 42, 112054. doi: 10.1016/j.celrep.2023.112054.

Reviewer 2 Report

Comments and Suggestions for Authors

Title: Intranasal immunisation of pneumococcal pep27 mutant attenuates allergic and inflammatory diseases by upregulating skin and mucosal Tregs

I have a few comments:

Line 112: 2.5µl should be 2.5µL

Line 137: Authors can add information regarding the genus Gemmiger, which is associated with the deconjugation of glycine-conjugated bile acids. Also, they can add the genera Eubacterium and Ruminococcus, which are linked to the deconjugation of taurocholic acid [information can be found in Martin et al. 2018].  

Section 7 Intranasal immunisation of Δpep27 protects allergic diseases: Authors should provide information concerning the mediation of IL-27 by Tregs and its impact on mitigating allergic lung inflammation and airway hyperresponsiveness in asthma models [information can be found in Suzuki et al 2019, et al, Lu 2020].  The authors may also include the role of IL-27 in airway remodelling and epithelial-mesenchymal transition in asthma via the modulation of the STAT1 and STAT3 pathways and the downregulation of the RhoA/ROCK signalling pathway [information can be found in Liu et al. 2019, Huang et al. 2022].

Author Response

Reviewer 2

Title: Intranasal immunization of pneumococcal pep27 mutant attenuates allergic and inflammatory diseases by upregulating skin and mucosal Tregs

I have a few comments:

Line 112: 2.5µl should be 2.5µL

Answer: Thank you for your valuable feedback. We have revised it:

Line 137: Authors can add information regarding the genus Gemmiger, which is associated with the deconjugation of glycine-conjugated bile acids. Also, they can add the genera Eubacterium and Ruminococcus, which are linked to the deconjugation of taurocholic acid [information can be found in Martin et al. 2018].

Answer: Thank you for your valuable feedback. We have added the followings to the revised manuscript:

“In addition, deconjugation of glycine-conjugated bile acids is associated with the genus Gemmiger, and deconjugation of taurocholic acid is linked to the genera Eubacterium and Ruminococcus [Martin G et al., 2018].”

Section 7 Intranasal immunisation of Δpep27 protects allergic diseases: Authors should provide information concerning the mediation of IL-27 by Tregs and its impact on mitigating allergic lung inflammation and airway hyperresponsiveness in asthma models [information can be found in Suzuki et al 2019, et al, Lu 2020].  The authors may also include the role of IL-27 in airway remodelling and epithelial-mesenchymal transition in asthma via the modulation of the STAT1 and STAT3 pathways and the downregulation of the RhoA/ROCK signalling pathway [information can be found in Liu et al. 2019, Huang et al. 2022].

Answer: Thank you for your valuable feedback. We have added the followings to the revised manuscript:

“In asthma models, IL-27 is an anti-inflammatory cytokine that belongs to the IL-12 family and is primarily expressed on dendritic cells, macrophages, and monocytes [Branchett WJ, Lloyd CM, 2019]. Because IL-27 targets Tregs in asthma models [Nguyen QT et al, 2019] and reduces respiratory allergy symptoms [Lu D et al, 2020; Suzuki M et al, 2019], the asthma-relieving effects of pep27 appears to be due to suppression of inflammation by IL-27-induced Tregs.

In bronchial asthma, inflammation is frequently triggered repeatedly, resulting in increased microvascular permeability and edema, which eventually leads to airway remodeling and thickening of the airway walls [Khan MA, 2020]. Injection of Treg cells after the onset of chronic asthma disease prevented airway remodeling [Kearley J, et al, 2008], indicating that Tregs can resolve chronic inflammation in asthma models. Thus, Tregs induced by pep27 may also contribute to resolving airway remodeling.”

However, the reviewer's comments about the role of IL-27 in airway remodeling are not the same as pep27; therefore, we have added a discussion of this as follows;

“In contrast to Tregs, IL-27 appears to act differently in asthma suppression; When IL‑27 was administrated therapeutically, IL‑27 did not ameliorate airway inflammation, airway hyperresponsiveness, and airway remolding. Prophylactically, administration of IL-27 decreased the concentration of Th2 cytokines and increased the number of type 1 regulatory T (Tr1) cells in the lungs [Huang et al., 2022; Liu et al., 2019], suggesting that IL-27 may prevent asthma. Thus, they demonstrated that IL-27 may be useful for preventive purposes but not for therapeutical purposes. pep27, however, works as both a therapeutic and preventive method.”

References

Branchett, W.J.; Lloyd, C.M. Regulatory cytokine function in the respiratory tract. Mucosal Immunol. 2019, 12, 589-600. doi: 10.1038/s41385-019-0158-0.

Huang, C.; Sun, Y.; Liu, N.; Zhang, Z.; Wang, X.; Lu, D.; Zhou, L.; Zhang, C. IL-27 attenuates airway inflammation and epithelial-mesenchymal transition in allergic asthmatic mice possibly via the RhoA/ROCK signalling pathway. Eur Cytokine Netw. 2022, 33, 13-24. English. doi: 10.1684/ecn.2021.0476. PMID: 36102857.

Khan, M.A. Regulatory T cells mediated immunomodulation during asthma: a therapeutic standpoint. J Transl Med. 2020, 18, 456. doi: 10.1186/s12967-020-02632-1.

Kearley, J.; Robinson, D.S.; Lloyd, C.M. CD4+CD25+ regulatory T cells reverse established allergic airway inflammation and prevent airway remodeling. J Allergy Clin Immunol. 2008, 122, 617-24.e6. doi: 10.1016/j.jaci.2008.05.048.

Liu, X.; Li, S.; Jin, J.; Zhu, T.; Xu, K.; Liu, C.; Zeng, Y.; Mao, R.; Wang, X.; Chen, Z. Preventative tracheal administration of interleukin-27 attenuates allergic asthma by improving the lung Th1 microenvironment. J Cell Physiol. 2019, 234, 6642-6653. doi: 10.1002/jcp.27422.

Lu, D.; Lu, J.; Ji, X.; Ji, Y.; Zhang, Z.; Peng, H.; Sun, F.; Zhang, C. IL‑27 suppresses airway inflammation, hyperresponsiveness and remodeling via the STAT1 and STAT3 pathways in mice with allergic asthma. Int J Mol Med. 2020, 46, 641-652. doi: 10.3892/ijmm.2020.4622.

Martin, G.; Kolida, S.; Marchesi, J.R.; Want, E.; Sidaway, J.E.; Swann, J.R. In Vitro Modeling of Bile Acid Processing by the Human Fecal Microbiota. Front Microbiol. 2018, 9, 1153. doi: 10.3389/fmicb.2018.01153.

Nguyen, Q.T.; Jang, E.; Le, H.; Kim, S.; Kim, D.; Dvorina, N.; Aronica, M.A.; Baldwin, W.M. 3rd, Asosingh K, Comhair S, Min B. IL-27 targets Foxp3+ Tregs to mediate antiinflammatory functions during experimental allergic airway inflammation. JCI Insight. 2019, 4, e123216. doi: 10.1172/jci.insight.123216.

 Suzuki, M.; Yokota, M.; Ozaki, S.; Matsumoto, T. Intranasal Administration of IL-27 Ameliorates Nasal Allergic Responses and Symptoms. Int Arch Allergy Immunol. 2019, 178, 101-105. doi: 10.1159/000493398.

Reviewer 3 Report

Comments and Suggestions for Authors

The present manuscript, “Intranasal Immunization of Pneumococcal pep27 Mutant Attenuates Allergic and Inflammatory Diseases by Upregulating Skin and Mucosal Tregs,” describes the properties of Δpep27, a mucosal vaccine candidate, which could extend its use to treating diseases like IBD and allergies. The induction of Tregs via intranasal immunization is an important property for the immunotherapy of allergic and recurrent inflammatory diseases. This is a very interesting and novel area of research with the potential to offer solutions to challenging diseases.

Immunization with Δpep27 provides mucosal protection, particularly in the lungs and gut, and emphasizes the role of Tregs in establishing immune tolerance. It also induces M2 macrophages, an antioxidant environment to reduce the stress response, and Treg attenuated caspase-14 and Wnt5a expression independent of the inflammatory environment. This suggests a strong anti-inflammatory mucosal tolerance and subsequent restoration of the gut microbiota, ensuring that barrier integrity is maintained for intestinal immune homeostasis. The Δpep27 immunization appears to promote the development and restoration of functional Treg cells in the skin, internal and respiratory organs, presumably through mucosal Treg infiltration or induction. It may also have effects on the gut-liver and gut-brain axis, which could cooperate with the brain or liver to exert anti-inflammatory effects on the skin and other organs. Overall, Δpep27 may be a promising mucosal vaccine candidate therapy for clinical applications in allergic and inflammatory diseases.

The authors consider as well the areas of future studies. For example, to determine whether the pep27 vaccine also induces Tregs in the brain and liver. Limitations were taken into account as well. Thus, from my point of view, the article is scientifically sound, interesting, and the information given to the reader is really valuable and complete.

On the other side, the manuscript needs important improvements for a proper presentation of the information. It is important for the reader to receive information about the Δpep27 from the introduction, as this is the subject matter of the review. It is frustrating to read four sections without mentioning the Δpep27 - only one sentence in a small paragraph of the third section, in ref 42! This is a major problem and the authors should be encouraged to transform their manuscript to focus on the subject of the review. Maybe section 2 is too long. The information should be provided in connection to the properties of the product. In case this work is completed, I find the article would be accepted for publication due to the relevance and scientific value of the manuscript.

Comments on the Quality of English Language

In Conclusions, the authors should change the sentence starting: In this study ... by: In this review... and double check the rest of the sentence... maybe it was taken from the original article, but this manuscript is a Review article.

Author Response

Reviewer 3

The present manuscript, “Intranasal Immunization of Pneumococcal pep27 Mutant Attenuates Allergic and Inflammatory Diseases by Upregulating Skin and Mucosal Tregs,” describes the properties of Δpep27, a mucosal vaccine candidate, which could extend its use to treating diseases like IBD and allergies. The induction of Tregs via intranasal immunization is an important property for the immunotherapy of allergic and recurrent inflammatory diseases. This is a very interesting and novel area of research with the potential to offer solutions to challenging diseases.

Immunization with Δpep27 provides mucosal protection, particularly in the lungs and gut, and emphasizes the role of Tregs in establishing immune tolerance. It also induces M2 macrophages, an antioxidant environment to reduce the stress response, and Treg attenuated caspase-14 and Wnt5a expression independent of the inflammatory environment. This suggests a strong anti-inflammatory mucosal tolerance and subsequent restoration of the gut microbiota, ensuring that barrier integrity is maintained for intestinal immune homeostasis. The Δpep27 immunization appears to promote the development and restoration of functional Treg cells in the skin, internal and respiratory organs, presumably through mucosal Treg infiltration or induction. It may also have effects on the gut-liver and gut-brain axis, which could cooperate with the brain or liver to exert anti-inflammatory effects on the skin and other organs. Overall, Δpep27 may be a promising mucosal vaccine candidate therapy for clinical applications in allergic and inflammatory diseases.

The authors consider as well the areas of future studies. For example, to determine whether the pep27 vaccine also induces Tregs in the brain and liver. Limitations were taken into account as well. Thus, from my point of view, the article is scientifically sound, interesting, and the information given to the reader is really valuable and complete.

On the other side, the manuscript needs important improvements for a proper presentation of the information. It is important for the reader to receive information about the Δpep27 from the introduction, as this is the subject matter of the review. It is frustrating to read four sections without mentioning the Δpep27 - only one sentence in a small paragraph of the third section, in ref 42! This is a major problem and the authors should be encouraged to transform their manuscript to focus on the subject of the review. Maybe section 2 is too long. The information should be provided in connection to the properties of the product. In case this work is completed, I find the article would be accepted for publication due to the relevance and scientific value of the manuscript.

Ans. We agree with the reviewer comment. We also moved the introduction to the pep27 mutant to the second section to increase readability for the reader, and added a description of the defense mechanism and efficacy by Tregs in the lung on line 363 to better understand pep27 efficacy as follows;

“Tregs maintain homeostasis by suppressing excessive immune activation. Neutrophils play an important role in eliminating pathogens during respiratory infections, and when they are mobilized to the lungs where pathogens have invaded, they phagocytose the pathogens and then digest and kill them by producing reactive oxygen species. However, in bacterial pneumonia, this process can lead to excessive lung damage and respiratory failure. Therefore, neutrophil phagocytosis (efferocytosis), the phagocytic removal of dead or dying neutrophils, is a key process in resolving lung inflammation. Tregs interact with alveolar macrophages to promote neutrophil phagocytosis and promote recovery. Overall, Tregs resolve inflammation and orchestrate tissue protection and airway system repair in mice and humans [D'Alessio FR et al, 2009; Jovisic M, et al., 2023; Proto JD et al, 2018; Xu R et al, 2023]. After three intranasal vaccinations of Δpep27, fluorescence-activated cell sorting (FACS) analysis of splenocytes showed an increase in Treg expression proportional to the number of immunizations. Additionally, it was confirmed that Tregs were induced in serum and bronchoalveolar lavage fluid (BALF) (Kim GL, Manuscript in preparation). Thus, Tregs may be one of the mechanisms by which pathogen infection is defended against by intranasal Δpep27 immunization.”

However, keeping the theme of the manuscript in mind, we first reviewed the importance of the gut-lung-brain axis in mucosal diseases and later discussed the therapeutic Δpep27 efficacy in the management of these diseases, including future perspectives.

Comments on the Quality of English Language

In Conclusions, the authors should change the sentence starting: In this study ... by: In this review... and double check the rest of the sentence... maybe it was taken from the original article, but this manuscript is a Review article.

Ans. We omitted and rephrased the sentence using ‘in this study’.

References

D'Alessio, F.R.; Tsushima, K.; Aggarwal, N.R.; West, E.E.; Willett, M.H.; Britos, M.F.; Pipeling, M.R.; Brower, R.G.; Tuder, R.M.; McDyer, J.F.; King, L.S. CD4+CD25+Foxp3+ Tregs resolve experimental lung injury in mice and are present in humans with acute lung injury. J Clin Invest. 2009,119, 2898-913. doi: 10.1172/JCI36498.

Jovisic, M.; Mambetsariev, N.; Singer, B.D.; Morales-Nebreda, L. Differential roles of regulatory T cells in acute respiratory infections. J Clin Invest. 2023, 133, e170505. doi: 10.1172/JCI170505.

Proto, J.D.; Doran, A.C.; Gusarova, G.; Yurdagul, A. Jr.; Sozen, E.; Subramanian, M.; Islam, M.N.; Rymond, C.C.; Du, J.; Hook, J.; Kuriakose, G.; Bhattacharya, J.; Tabas I. Regulatory T Cells Promote Macrophage Efferocytosis during Inflammation Resolution. Immunity. 2018, 49, 666-677.e6. doi: 10.1016/j.immuni.2018.07.015.

Xu, R.; Jacques, L.C.; Khandaker, S.; Beentjes, D.; Leon-Rios, M.; Wei, X.; French, N.; Neill, D.R.; Kadioglu, A. TNFR2+ regulatory T cells protect against bacteremic pneumococcal pneumonia by suppressing IL-17A-producing γδ T cells in the lung. Cell Rep. 2023, 42, 112054. doi: 10.1016/j.celrep.2023.112054.
